# Toward Improved Detection of Cholesteatoma Recidivism: Exploring the Role of Non-EPI-DWI MRI

**DOI:** 10.3390/jcm13092587

**Published:** 2024-04-28

**Authors:** Natalia Díaz Zufiaurre, Marta Calvo-Imirizaldu, Joan Lorente-Piera, Pablo Domínguez-Echávarri, Pau Fontova Porta, Manuel Manrique, Raquel Manrique-Huarte

**Affiliations:** 1Otorhinolaryngology Department, University of Navarra Clinic, 31008 Pamplona, Spain; ndiazzu@unav.es (N.D.Z.); mmanrique@unav.es (M.M.); rmanrique@unav.es (R.M.-H.); 2Radiology Department, University of Navarra Clinic, 31008 Pamplona, Spain; mcalvoi@unav.es (M.C.-I.); pdaniel@unav.es (P.D.-E.); 3Faculty of Medicine, University of Navarra, 31008 Pamplona, Spain; pfontova@alumni.unav.es

**Keywords:** cholesteatoma, primary acquired cholesteatoma, attic exposure–antrum exclusion, non-echo-planar diffusion MRI

## Abstract

**Background:** Cholesteatoma is a lesion capable of destroying surrounding tissues, which may result in significant complications. Surgical resection is the only effective treatment; however, the presence of cholesteatoma recidivism is common. This study evaluated the effectiveness of the Attic Exposure-Antrum Exclusion (AE-AE) surgical technique in treating cholesteatomas and identifying factors associated with recidivism. Additionally, the study aimed to assess the utility of non-echo-planar diffusion MRI (non-EPI-DWI MRI) in detecting cholesteatoma recidivism in patients undergoing AE-AE surgery. **Methods:** The study involved 63 patients who underwent AE-AE surgery for primary acquired cholesteatoma and were followed up clinically and radiologically for at least five years. The radiological follow-up included a non-EPI-DWI MRI. **Results:** Results showed that the AE-AE technique successfully treated cholesteatomas, with a recidivism rate of 5.2%. The study also found that non-EPI-DWI MRI was a useful diagnostic tool for detecting cholesteatoma recidivism, although false positives could occur due to the technique’s high sensitivity. As Preoperative Pure-tone average (PTA) increases, there is a higher probability of cholesteatoma recidivism in imaging tests (*p* = 0.003). **Conclusions:** Overall, the study highlights the importance of the AE-AE surgical technique and non-EPI-DWI MRI in managing cholesteatoma recidivism in patients, providing valuable insights into associated risk factors and how to manage recidivism. Non-EPI-DWI MRI can assist in patient selection for revision surgery, reducing unnecessary interventions and associated risks while improving treatment outcomes and patient care.

## 1. Introduction

Cholesteatomas are considered epidermal inclusion cysts that originate within the temporal bone, consisting of squamous epithelial cells and related debris. These growths tend to enlarge progressively, leading to tissue destruction, frequently accompanied by inflammation and granulation tissue formation [1].

Cholesteatomas are typically categorized into three groups: congenital, primary acquired, and secondary acquired. Hypotheses suggest that keratinized squamous epithelial tissue in the middle ear cleft gives rise to congenital cholesteatomas. The primary location of RP often occurs in the pars flaccida, a region commonly associated with the most frequently diagnosed syndrome among symptomatic patients. Eustachian tube dysfunction (ETD) and gradual thinning of the tympanic membrane frequently coincide with RP, alongside recurrent occurrences of otitis media with effusion [2]. Secondary acquired cholesteatomas, on the other hand, result from tympanic membrane perforations with epithelial migration [3]. Several factors, including tobacco use, previous infections, and genetic predispositions, may also contribute to the development of cholesteatomas [4,5].

Cholesteatomas possess the capacity to induce recurrent infections and osseous erosion, which can affect diverse structures within the temporal bone. Consequently, this can lead to auditory impairment, facial nerve compromise, vestibular dysfunction, and potentially intracranial sequelae [1].

Various treatment options for retraction pockets are available, ranging from conservative approaches to surgical interventions [6]. For this condition, the insertion of ventilation tubes, excision of RP with tympanic membrane reconstruction, and tympanoplasty with canal wall-up or canal wall-down procedures can be performed depending upon the case. The lack of consensus stems from the challenge of predicting which instances of RP will progress into cholesteatoma and which will remain stable and symptom-free [2].

Canal wall-down procedures are preferred for complete cholesteatoma removal, while canal wall-up procedures preserve the middle ear’s anatomy and offer better reconstruction options [7].

One of the significant challenges associated with cholesteatomas is their tendency to recur. Consequently, close follow-up of surgically treated patients is necessary [8]. Early diagnosis of cholesteatoma recidivism is crucial in preventing complications [9]. Traditionally, a second surgery was scheduled 6 months to 1 year after the initial surgery to check for recurrence. However, there was uncertainty about when and how to perform this second procedure. Cholesteatoma recurrence is often linked to pre-existing chronic conditions or when some of the cholesteatoma remains unintentionally or intentionally due to its proximity to important structures [10]. Initially, High-resolution Computed Tomography (CT) scans were the preferred way to diagnose and assess the extent of cholesteatoma. However, they had limitations in distinguishing between fibrotic tissue, cholesteatoma, or inflammation. Later, Magnetic Resonance diffusion-weighted imaging with echo-planar sequences (EPI-DWI MRI) was employed for lesions larger than 5 mm. Nevertheless, its effectiveness was limited when dealing with smaller lesions, leading to the generation of several artifacts. The development of Magnetic Resonance diffusion-weighted imaging with non-echo-planar sequences (non-EPI-DWI MRI) marked a significant breakthrough in imaging for detecting small cholesteatomas. This technique may help diagnose cholesteatoma recidivism after the first surgery, potentially avoiding unnecessary revision surgeries and associated risks. 

### Objectives

The objective of this study is to measure the rate of cholesteatoma recidivism after a specific surgical technique, Attic Exposure-Antrum Exclusion (AE-AE), and the role of non-EPI-DWI MRI sequences for the detection of cholesteatoma recidivism. Additionally, it seeks to investigate different parameters that may contribute to its recurrence. 

## 2. Materials and Methods

### 2.1. Study Design

A retrospective cohort study was conducted at the Department of Otorhinolaryngology in a tertiary-level hospital from 2003 to 2021. The study included 63 patients who underwent Attic Exposure and Antrum Exclusion (AE-AE) surgery for primary acquired cholesteatoma. Patients were followed up clinically by otoscopy and audiometry and radiologically by MRI with a diffusion sequence (non-EPI DWI MRI) two years after surgery to check for the presence of a cholesteatoma recidivism.

### 2.2. Inclusion and Exclusion Criteria

All patients underwent cholesteatoma surgery with the AE-AE technique and were followed-up with non-EPI-DWI MRI. 

Approval from the Institutional Review Board and informed consent from all patients were obtained (RGPD 2016/679). Age was not a requirement for inclusion in the study, as patients ranged in age from 6 to 86 years. Additionally, the cholesteatoma could not exceed the aditus, or if it did, it could not damage any intracranial structures or the labyrinth. Patients who did not undergo non-EPI DWI MRI as a follow-up imaging technique for particular reasons were not included.

### 2.3. Diffusion MRI with Non-Echo-Planar Sequences (Non-EPI-DWI MRI)

The imaging modality used in this study is Diffusion MRI with non-echo-planar sequences (non-EPI DWI MRI). DWI-MRI hinges on the fundamental concept of diffusion, which entails the random movement of free water molecules within tissue. When this movement is reduced from what we would expect for that tissue, diffusion is “restricted”. In DWI-MRI, tissues exhibiting restricted diffusion manifest as hyperintense areas, whereas those where diffusion is facilitated appear as hypointense regions. In the case of cholesteatoma, the buildup of keratin limits the free movement of water molecules, resulting in the lesion appearing as hyperintense [9]. 

The study used non-echo-planar sequences, specifically “half-Fourier single-shot turbo spin echo” (HASTE) sequences, which offer superior image quality characterized by improved spatial resolution and reduced susceptibility to artifacts. The study utilized a 3 Tesla Magnet, Siemens Magnetom Vida (Siemens Healthineers, Erlangen, Germany) for the MRI scans [11]. The presence of cholesteatoma was evaluated at our center by the radiologists according to the DWI signal intensity relative to that of the white matter. A few patients underwent multiple examinations, resulting in the analysis of several hyperintense DWI focuses analyzed in their cases. The rate of growth was determined by either measuring the duration between the definitive surgery and the initial DWI-positive study or assessing the size alteration observed across successive positive MRI examinations.

### 2.4. Patient Follow-Up 

After the surgical intervention, a first dressing change was performed 10 days later. Subsequently, a second postoperative visit was conducted one month after the surgery, followed by periodic revisions every 1–2 months for an initial period of 5–6 months, and then annually for monitoring purposes. A follow-up period of at least 5 years was required, during which the first imaging test was performed two years after the surgery, provided there were no signs of suspicion before this period.

### 2.5. European Academy of Otology and Neuro-Otology (EAONO) Staging System

Regarding the extent of cholesteatoma involvement, the STAM system was used to divide the middle ear into 5 spaces: S1 (protympanum), T (tympanic cavity), A (attic), M (mastoid), and S2 (sinus tympani). The EAONO staging system was used to measure the extent of involvement, which is divided into 4 stages depending on whether it is localized to the primary site (stage 1), invades two or more sites (stage 2), presents with extracranial complications (stage 3), or presents with intracranial complications (stage 4) [12].

### 2.6. Pure-Tone Average (PTA)

The patient’s hearing capacity at speech frequencies was assessed using the Pure-tone average (PTA). This metric was computed using the mean of the air conduction hearing thresholds at 500, 1000, 2000, and 4000 Hz, derived from a tonal audiogram test conducted within an audiometric chamber (Audiotest, Equinox IEC 645-1/ANSI 53.6-1996 type I, IEC 645-2/ANSI S3.6-1996 type B, Assens, Denmark). Patients underwent the test before and after the intervention. During the test, the patient wears headphones and signals when they hear pure tones transmitted at specific frequencies (hertz) and volumes (decibels) in each ear. The results were recorded and used to create a graph of the minimum volume required to hear. A bone oscillator was also used to evaluate bone conduction, and the results were recorded on the same graph [13].

### 2.7. Surgical Technique

In the study, patients underwent surgery using the AE-AE technique. The AE-AE procedure is classified as a canal wall-down technique, offering additional benefits. This method exposes the attic by creating an opening in the superior wall of the external auditory canal with a drill. It further involves isolating the antrum and mastoid cells through aditus closure using a graft. It offers several benefits, such as achieving precise management of the frequently affected attic areas by cholesteatoma, thereby diminishing the likelihood of recidivism while maintaining the potential for restoring hearing function. The exclusion of the antrum effectively seals off the mastoid spaces. This closure minimizes the exposed area, promotes self-cleaning, and permits water entry without causing dizziness. The AE-AE approach is recommended when the lesion remains confined within the aditus or, in the case that it extends beyond when it is a localized cholesteatoma that does not damage the labyrinth [7]. After the excision was performed, mesotympanic inspection with a 30- and 45-degree angled endoscope was carried out only in those patients with doubts regarding whether it had been complete. Patients who did not meet the criteria for this technique underwent a modified radical mastoidectomy or a radical mastoidectomy.

### 2.8. Statistical Analysis

Regarding statistical analysis, a retrospective cohort study was conducted on patients who underwent non-EPI-DWI MRI as an imaging test in the follow-up after cholesteatoma removal with the AE-AE technique. The study utilized the SPSS IBM version 20.0 software. Understanding the importance of time in the development and recurrence of cholesteatoma, in our study, a statistical analysis was conducted using logistic regression to investigate the association between the occurrence of recidivism and various parameters, including EAONO classification, pure tone audiometry (PTA), surgical technique utilized, type of reconstruction for aditus closure and ossicular chain, as well as findings from non-EPI-DWI MRI. While the study design was not prospective, measures were taken to control for the time factor in the statistical analysis, including the use of appropriate logistic regression models capable of handling time variables as covariates. Additionally, robust statistical methods were employed to ensure that the potential effects of time on cholesteatoma recidivism were adequately considered.

## 3. Results

During radiologic follow-up, all 63 (100%) patients underwent non-EPI-DWI MRI to detect the presence of cholesteatoma recidivism. The study population consisted of 51% females and 49% males, with a mean age of 41 (standard deviation: 21) at the time of surgery. Among these patients, 57% had cholesteatoma on the left side, while the remaining 43% were on the right side. The analysis of the parameters in this patient cohort, using non-EPI-DWI MRI as the imaging follow-up technique, yielded the following results (Table 1).

According to the European Academy of Otology and Neuro-Otology (EAONO) and Japanese Otological Society (JOS) classification for cholesteatoma, we found 47.6% of patients in Stage I (cholesteatoma localized in the attic), 47.6% in Stage II (cholesteatoma involving two or more sites), and 4.8% in Stage III (cholesteatoma with extracranial complications such as facial palsy, labyrinthine fistula, or canal wall destruction). No patient was found with Stage IV (cholesteatoma with intracranial complications).

About the surgical technique used, 67% of patients underwent a simple attic exposure and antrum exclusion (AE-AE) technique, 22% underwent an extended attic exposure and antrum exclusion (AE-AE) technique, and 11% underwent a radical mastoidectomy.

For aditus closure, cartilage was used in 90% of cases; preferably, conchal cartilage was utilized, thus avoiding obtaining it from the tragus, which could compromise its supporting function in a potential need for hearing aid use. Cortical bone was used in in 5%, and a combination of cartilage and cortical bone in 5% of cases.

Regarding reconstruction of the ossicular chain, in 60% of cases, reconstruction was unnecessary because the cholesteatoma was located laterally and did not affect the integrity and mobility of the ossicular chain. In 22% of cases, the stapes suprastructure was preserved, and a partial ossicular chain prosthesis (PORP) was placed. However, in 18% of cases, the stapes suprastructure was absent, and a total ossicular chain prosthesis (TORP) was needed.

Regarding the pure-tone average (PTA), the pre-surgical PTA was 42.2 dB (standard deviation: 15.7) and the post-surgical PTA at 12 months after surgery was 40.9 dB (standard deviation: 15.9). The difference between the 12-month post-surgical PTA and the pre-surgical PTA was −1.3 dB (standard deviation: 13.0). The *p*-value was 0.457, as shown in Table 2.

Statistical analyses using logistic regression showed no statistically significant association between cholesteatoma recidivism and any of the parameters examined in Table 3.

However, logistic regression analysis revealed a significant association between pre-surgical PTA and cholesteatoma recidivism on non-EPI-DWI MRI scans (*p*-value: 0.003).

So, in Table 3, where logistic regression was used, the coefficient values indicate the relative contribution of each independent variable to predict cholesteatoma recidivism. A significant coefficient value suggests that the variable is associated with the probability of recidivism, regardless of other variables in the model. In this context, the value of 0.185 for Pre-Surgical PTA indicates that, after controlling for other variables in the model, an increase in PTA is associated with a change in the probability of recidivism.

On the other hand, when we evaluate the relationship between imaging findings and preoperative audiometry using logistic regression. Here, the *p*-value of 0.003 indicates a statistically significant association between PTA and imaging findings. This suggests that PTA has an impact on imaging outcomes, which could have significant clinical implications, such as the ability to predict cholesteatoma recidivism.

In summary, even if it may seem contradictory on the PTA results, both analyses may be providing valuable information but are focused on different aspects of the study. Table 3 examines variables predicting cholesteatoma recidivism, while the significant *p*-value of 0.003 explores the relationship between PTA and imaging findings.

The results obtained from the non-EPI-DWI MRI scans revealed that 87% (55 patients) did not show any signs of cholesteatoma recidivism. However, in 13% (8 patients), the imaging results suggested the possibility of cholesteatoma recidivism. Of these 8 patients, one individual underwent a second non-EPI-DWI MRI scan, which yielded negative results, suggesting a false positive in the initial scan. The remaining 7 patients were scheduled for a second look surgery. Unfortunately, 2 cases were lost to follow-up, but the remaining 5 patients underwent the surgical intervention. Among these operated patients, 4 cases were found to have a cholesteatoma recidivism stage II due to multiple locations, which was successfully removed. However, in one of the operated cases, it turned out to be another false positive as no cholesteatoma was found during the intervention.

Additionally, three of these patients required a third revision surgery. All three patients had already undergone the surgical revision. In one case, the positive diffusion MRI result corresponded to a cholesterol granuloma; in the second case, cholesteatoma was found during both revision surgeries, and in the third case, inflammatory tissue was detected. The results are summarized in Figure 1.

Given the observed results, the values of non-EPI-DWI MRI for detecting cholesteatoma recidivism in terms of sensitivity, specificity, positive predictive value, and negative predictive value were 100% (CI: 100.0–100.0%), 96.4%(CI: 91.5–100.0%), 60% (CI: 47.4–72.6%), and 100% (CI: 100.0–100.0%), respectively. As for the surgical technique of AE-AE, it was successful, with a recidivism rate of 5.2% (CI: 0.0–10.87%).

## 4. Discussion

Surgical resection stands as the primary treatment approach for cholesteatoma, with a spectrum of surgical techniques available, each tailored to specific considerations. These techniques include tympanotomy, atticotomy, cortical mastoidectomy, and canal wall up or down mastoidectomy, as well as more extensive interventions such as modified–radical and radical mastoidectomy. The overarching aim remains the complete eradication of the cholesteatoma while striving to reduce the potential for recidivism, conserve auditory function, and improve ear hygiene. The choice of surgical approach is contingent upon the cholesteatoma’s extent and the surgeon’s experience. In contrast, atticotomy is preferred when the cholesteatoma is confined to the lateral side of the malleus and incus within the “attic”. Success rates for complete cholesteatoma removal with atticotomy typically fall between 70% and 90%. Cortical mastoidectomy comes into play when the cholesteatoma extends towards the ossicles and penetrates the mastoid region via the antrum. This approach typically yields success rates ranging from 70% to 95%. The selection between canal wall-up and canal wall-down procedures depends on the attainability of complete cholesteatoma elimination. Opting for canal wall-down procedures has been demonstrated to offer enhancements in postoperative physical examination outcomes and improved surgical visualization. On the other hand, selecting canal wall-up procedures allows for the preservation of the natural middle ear anatomy, thereby facilitating improved reconstruction capabilities. This results in a middle ear that closely mimics its physiological state, necessitates less postoperative care, improves the compatibility with hearing aids, and the absence of water-related restrictions [10]. The reported success rates for complete cholesteatoma removal with canal wall-up mastoidectomy typically range from 70% to 90%. Canal wall-down mastoidectomy demonstrates success rates of 80% to 95%. More extensive procedures, such as modified-radical and radical mastoidectomy, yield success rates ranging from 85% to 95% [14,15,16,17,18].

The surgical technique of AE-AE has shown to be an effective technique for the resection of cholesteatoma, with a recidivism rate of 5.2% in our study population. Another study tracked 42 patients for 6 months to 7 years after their initial cholesteatoma surgery using the same technique. They found a recidivism rate of 4.8% [7]. The lower rate in this study might be because they used less sensitive imaging like CT scans, which may have missed smaller cholesteatomas that MRI could have detected. Additionally, our study included patients with more aggressive cholesteatomas that affected the ossicular chain, which is associated with a higher recidivism rate.

Regarding imaging and postoperative monitoring, high-resolution CT scans were widely used as the main imaging technique for diagnosing and characterizing cholesteatoma. However, its ability to accurately distinguish between fibrotic tissue, cholesteatoma, and inflammation is limited. The interpretation of temporal bone CT entails a labor-intensive process due to the intricate anatomical structures and microarchitecture involved, heavily relying on the reader’s expertise [19]. Later, EPI-DWI MRI was introduced as a mean to identify cholesteatoma recidivism. Nevertheless, its efficacy in detecting smaller lesions, measuring less than 5 mm, was hindered by magnetic field inhomogeneity artifacts, posing challenges for accurate interpretation. The emergence of non-EPI-DWI MRI has significantly advanced imaging techniques specifically tailored for detecting small cholesteatomas. This method enhances both sensitivity and specificity, as it minimizes artifacts in the temporal bone region and provides superior spatial resolution. Traditional morphological MRI sequences, weighted in T1 and T2, though widely used, exhibit limited specificity in diagnosing inflammatory conditions of the middle ear. The characteristics of cholesteatoma, granulation tissue, or chronic otitis media can manifest differently in these sequences, resulting in a lack of consistent reliability when attempting to differentiate between these conditions [9].

Diffusion MRI is a highly sensitive and specific technique. This enables the selection of patients for revision surgery, avoiding unnecessary interventions and associated risks. The sensitivity (100%), specificity (96.4%), positive predictive value (60%), and negative predictive value (100%) obtained in our study with this technique are similar to those usually reported.

It is essential to highlight that non-EPI-DWI MRI can occasionally yield false-positive results if another tissue with restricted diffusion is present. Ordinary postsurgical inflammatory alterations do not impede diffusion, but in rare instances, different tissues may develop such behavior. Common factors contributing to false-positive findings are cholesterol granuloma, purulent content, or an abscess in the middle ear. Conversely, cholesteatomas measuring less than 2 mm are frequently reported as the main culprits behind false-negative outcomes [9].

Our study’s results are consistent with those obtained in another study that evaluated non-EPI DWI MRI’s efficacy. The follow-up study included 35 patients who had undergone initial surgery for cholesteatoma resection, and non-EPI-DWI MRI was used as the diagnostic method. This technique detected 9 out of 10 cholesteatoma recurrences, with the only missed lesion being a 2 mm cholesteatoma with imaging affected by motion artifact in a child. These findings highlight the superior performance of non-EPI-DWI MRI compared to other imaging techniques [20,21]. High-resolution CT scans have shown limited reliability in detecting cholesteatoma recurrence, with a sensitivity of 43% and specificity of 48% in similar cases [8]. Another study involving 45 patients found that EPI-DWI MRI conducted a few months after initial cholesteatoma surgery had a sensitivity of only 12.5% [22]. These results underscore the effectiveness of non-EPI DWI MRI, which can accurately detect cholesteatoma recurrence with high sensitivity and specificity. However, also in terms of follow-up, recent studies such as that of Covelli et al. in 2022 [23], advised the possibility of extending follow-up to 7 years to obtain a better perspective and characterize tumor monitoring, if possible. Nevertheless, the authors, in accordance with our work, after analyzing their series of 64 patients, concluded that extending the follow-up to at least 5 years after primary surgery was also recommended to detect cholesteatoma recidivism beyond this time frame. Additionally, they also raised the possibility of performing an early imaging test, just one month after the procedure, to detect the presence of any relevant cholesteatomatous residue, thus suggesting a potential therapeutic failure and the possibility of carrying out an early and minimally invasive reintervention [23].

However, it is worth noting that sensitivity in another recent study was, unlike ours, 59%. This study included 33 patients who underwent a non-EPI-DWI MRI and a second revision surgery after the first intervention [24]. Nevertheless, several factors that may have influenced this discrepancy in results need to be evaluated. The number of false negative Non-EPI-DWI MRIs is not known since small recidivism in cholesteatoma can be asymptomatic for some years, this situation makes even harder some parameters to be calculated and it can call into question the role of Non-EPI-DWI MRI in some circumstances as a tool for detecting cholesteatoma recidivism. It is for this reason that a second MRI, i.e., 5 years after surgical intervention may be an alternative to a second look.

The initial aspect to consider is the duration between the imaging examination and the second surgical procedure. In most patients, there was a gap of two months between the two tests, but some patients had to wait from 188 to 201 days until the second surgical intervention was performed. The problem with long periods between both tests is that, given the high capacity of recidivism of this pathology, a new cholesteatoma may appear that did not exist at the time of the imaging test. Another factor is the waiting time between the first surgery and the non-EPI DWI MRI. In this study, approximately nine months elapsed; in ours, it was two years. This implies that the lesions found at nine months may be significantly smaller than those found at two years, making their diagnosis more challenging and increasing the number of false negatives.

Some studies consider the lack of surgical re-intervention as a limitation after not detecting any indication of recidivism in the non-EPI DWI MRI since they consider it the only way to confirm true negative results. Nonetheless, the patients in our study have been followed up through otoscopic examinations and radiological tests. Therefore, along with the high sensitivity of the imaging test, negative results can be classified as true negatives with a high level of certainty. In cases of recidivism MRI is not useful unless intracranial complication are suspected.

A limitation of our study is the heterogeneity among the cases included regarding the aggressiveness of the cholesteatoma. Based on the EAONO classification explained earlier, the study includes cases from the first, second, and third stages. This could lead to an overestimation of the aggressiveness of the pathology and its capacity of recidivism. One possible solution to this problem would be to add only those patients classified within the same EAONO stage as an inclusion criterion. Another possible alternative would be to increase the number of patients, allowing for the stratification of cases into different stages within the same study.

While our study did not identify any parameters demonstrating a statistically significant association with an increased risk of new cholesteatoma recidivism, our investigation did reveal a notable finding regarding the correlation between pre-surgical Pure Tone Average (PTA) and the likelihood of detecting cholesteatoma recidivism in non-EPI-DWI MRI scans (*p*-value: 0.003). This correlation indicates that as the PTA increases, there is a higher probability of detecting cholesteatoma in imaging tests. This finding underscores the critical importance of early diagnosis and treatment of this pathology before it infiltrates additional structures and exacerbates the progression of the disease. In summary, our study suggests that while certain parameters may not directly predict cholesteatoma recidivism, monitoring PTA levels can provide valuable insights into the likelihood of recidivism, aiding in timely intervention and management strategies.

To conclude future studies could optimize radiological techniques for earlier diagnosis of cholesteatoma recidivism. To address these gaps in knowledge and advance our understanding of this disease, it’s crucial to prioritize and support future research in this area.

## 5. Conclusions

Our study demonstrates that the AE-AE surgical technique effectively treats cholesteatoma with a low recidivism rate. Also, non-EPI-DWI MRI is a useful diagnostic tool for detecting cholesteatoma recidivism in patients undergoing AE-AE surgery. It is important to note that while this technique has high sensitivity, it may occasionally lead to false positive results.

The findings of our study emphasize the importance of using non-EPI-DWI MRI in the post-surgical management of cholesteatoma patients, as it allows an accurate patient selection for revision surgery and reduces the need of unnecessary interventions and associated risks. Additionally, our results suggest that as preoperative PTA value increases, the likelihood of detecting cholesteatoma recidivism in imaging tests also increases.

## Figures and Tables

**Figure 1 jcm-13-02587-f001:**
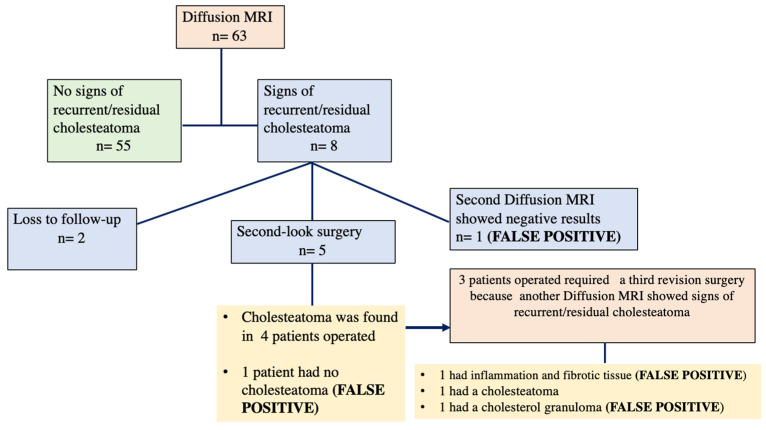
Patient follow-up scheme.

**Table 1 jcm-13-02587-t001:** Summary of the results referring to the EAONO classification, along with the surgical technique employed, exclusion of the aditus, and reconstruction of the ossicular chain, including their relevant percentages.

Eaono Classification	Percentage (%)
Stage I (Attic)	47.6%
Stage II (Multiple Sites)	47.6%
Stage III (Extracranial Complications)	4.8%
Stage IV (Intracranial Complications)	0%
**Surgical Technique**	**Percentage (%)**
Simple AE-AE	67%
Extended AE-AE	22%
Radical Mastoidectomy	11%
**Aditus Closure**	**Percentage (%)**
Cartilage	90%
Cortical Bone	5%
Cartilage + Cortical Bone	5%
**Ossicular Chain Reconstruction**	**Percentage (%)**
Not Necessary	60%
Partial Ossicular Chain Prosthesis (PORP)	22%
Total Ossicular Chain Prosthesis (TORP)	18%

**Table 2 jcm-13-02587-t002:** Results of the PTA in the follow-up period from pre-intervention to 6 months later, including the difference between them and the no statistical significance result.

Pure-Tone Average (PTA)	Pre-Surgical PTA (dB)	12 Month Post-Surgical PTA (dB)	Difference (dB)
Mean	42.2	40.9	−1.3
Standard Deviation	15.7	15.9	13

*p*-value: 0.457.

**Table 3 jcm-13-02587-t003:** Statistical analyses using logistic regression of possible factors associated with cholesteatoma recidivism using *p* and coefficient values. OR: Odds Ratio; CI: Confidence Interval.

Variables	Statistical Significance (*p* Value)	Coefficient Values (OR and CI)
Pre-surgical PTA	0.185	1.03 (0.98; 1.08)
Age	0.416	0.98 (0.93; 1.03)
Surgical technique	0.504	0.52 (0.08; 3.53)
Aditus closure	1.000	1.00 (0.04; 22.93)
Ossicular chain reconstruction	0.332	0.42 (0.07; 2.45)
EANO Classification	0.529	1.70 (0.33; 8.87)
Sex	0.943	0.93 (0.12; 7.08)

## Data Availability

Data pertaining to this study can be shared upon request to the corresponding author.

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
