# Peer review of "Toward Improved Detection of Cholesteatoma Recidivism: Exploring the Role of Non-EPI-DWI MRI"

_jcm, 2024, doi:10.3390/jcm13092587_

Round 1

Reviewer 1 Report

Comments and Suggestions for Authors

A very interesting and useful study regarding the effectiveness of the Attic Exposure - Antrum Exclusion surgical technique in treating cholesteatomas, in identification of the factors associated with recurrence and the utility of non-echo-planar diffusion MRI in cholesteatoma recurrence identification.

Author Response

A very interesting and useful study regarding the effectiveness of the Attic Exposure - Antrum Exclusion surgical technique in treating cholesteatomas, in identification of the factors associated with recurrence and the utility of non-echo-planar diffusion MRI in cholesteatoma recurrence identification.--> Thank you very much for your comment and for summarizing the main idea and objective of our work so well. We hope that the publication moves forward and proves to be truly helpful to other professionals, both surgeons and radiologists.

Reviewer 2 Report

Comments and Suggestions for Authors

good work. About the introduction I suggest a brief presentation of causes of cholesteatoma and its possible surgical approach. About this, you should mention this manuscript:

Immordino A, Salvago P, Sireci F, Lorusso F, Immordino P, Saguto D, Martines F, Gallina S, Dispenza F. Mastoidectomy in surgical procedures to treat retraction pockets: a single-center experience and review of the literature. Eur Arch Otorhinolaryngol. 2023 Mar;280(3):1081-1087. 

Author Response

good work. About the introduction I suggest a brief presentation of causes of cholesteatoma and its possible surgical approach: Thank you very much for your suggestion and for providing us with an interesting review in the bibliography for reference. Regarding the etiopathogenic section, you will see that we have attached in line 38: The primary location of RP often occurs in the posterior-superior quadrant of the pars tensa, a region commonly associated with the most frequently diagnosed syndrome among symptomatic patients. Eustachian tube dysfunction (ETD) and gradual thinning of the tympanic membrane frequently coincide with RP, alongside recurrent occurrences of otitis media with effusion.''

Evenmore, about the treatment, in line 51: Various treatment options are available for this condition, ranging from conservative approaches to surgical interventions [5]. However, The most effective treatment for cholesteatomas is surgical resection, which can be performed through various approaches, such as the insertion of ventilation tubes, excision of RP with tympanic membrane reconstruction, and tympanoplasty with canal wall-up or canal wall-down procedures. The lack of consensus stems from the challenge of predicting which instances of RP will progress into cholesteatoma and which will remain stable and symptom-free [2]''.

Reviewer 3 Report

Comments and Suggestions for Authors

The work shows numerous interesting parts. It might be useful to hypothesize carrying out a radiological evaluation after surgery one month later which could identify a residue early to plan a minimally invasive surgical procedure early

The authors recommend carrying out a final radiological check 5 years later, in a work in 2022 Covelli et al recommend regardless of the type of technique to carry out a radiological evaluation up to 7 years later, it could be useful to include this bibliographic reference :

Proposal of a magnetic resonance imaging follow-up protocol after cholesteatoma surgery: a prospective study Acta Oto-LaryngologicaVolume 142, Issue 6, Pages 484 – 490 2022.

It would be useful to know if the authors have used endoscopy in some cases to highlight the parts of the posterior mesotympanum? Were the patients with recurrent residual part of stage II or III? Could this be due to multiple locations or the presence of complications?

The authors describe the use of cartilage in the reconstruction, did they use that of the concha or tragus, for the reconstruction of the posterior wall or the mastoid filling?

Comments on the Quality of English Language

the Quality of English it is good

Author Response

1- The work shows numerous interesting parts. It might be useful to hypothesize carrying out a radiological evaluation after surgery one month later which could identify a residue early to plan a minimally invasive surgical procedure early. The authors recommend carrying out a final radiological check 5 years later, in a work in 2022 Covelli et al recommend regardless of the type of technique to carry out a radiological evaluation up to 7 years later, it could be useful to include this bibliographic reference :Thank you very much for your contribution and for citing an interesting article, with a similar sample size and a contribution to consider regarding follow-up and management. We have incorporated both comments into line 376 as follows: However, also in terms of follow-up, recent studies such as that of Covelli et al. in 2022, advised the possibility of extending follow-up to 7 years to obtain a better perspective and characterize tumor monitoring, if possible. Nevertheless, the authors, in accordance with our work, after analyzing their series of 64 patients, concluded that extending the follow-up to at least 5 years after primary surgery was also recommended to detect any recurrent cholesteatoma beyond this time frame. Additionally, they also raised the possibility of performing an early imaging test, just one month after the procedure, to detect the presence of any relevant cholesteatomatous residue, thus suggesting a potential therapeutic failure and the possibility of carrying out an early and minimally invasive reintervention. [22]"

2- It would be useful to know if the authors have used endoscopy in some cases to highlight the parts of the posterior mesotympanum? Were the patients with recurrent residual part of stage II or III? Could this be due to multiple locations or the presence of complications?--> Thank you for your input. As suggested, we have reviewed the cases and completed the information as follows in line 267:Among these operated patients, 4 cases were found to have a recurrent or residual cholesteatoma stage II due to multiple locations, which was successfully removed. However, in one of the operated cases, it turned out to be another false positive as no cholesteatoma was found during the intervention."

3- The authors describe the use of cartilage in the reconstruction, did they use that of the concha or tragus, for the reconstruction of the posterior wall or the mastoid filling?->Thank you again for your response. We have included not only the type of cartilage used in these procedures but also the justification for why we chose conchal cartilage. This is reflected in line 205: ‘’For aditus closure, cartilage was used in 90% of cases, with conchal cartilage being the preferred choice, thus avoiding obtaining it from the tragus, which could compromise its supporting function in a potential need for hearing aid use. Cortical bone was used in 5% of cases, and a combination of cartilage and cortical bone in 5% of cases."

Reviewer 4 Report

Comments and Suggestions for Authors

It seems a manuscript with the aim to promote the results of the surgical technique rather then to show the role of imaging because there is nothing new on cholesteatoma imaging, so the title is misleading

description of surgical technique should be in material and methods

recurrent cholesteatoma is time dependent, so the statistical analysis should consider time.

recurrent cholesteatoma is a clinical observation, imaging is useful for residual disease

table 3: title of column 2 is p value, but coefficient values are reported

Author Response

It seems a manuscript with the aim to promote the results of the surgical technique rather then to show the role of imaging because there is nothing new on cholesteatoma imaging, so the title is misleadingà-->Thank you very much for your feedback. We agree with your point and have modified it, providing another approach that emphasizes the role of diagnostic imaging, thus avoiding leading to a conclusion. As you can see, the new title is: "Toward improved detection of cholesteatoma recurrences: Exploring the role of Non-EPI-DWI MRI."

description of surgical technique should be in material and methods-> We agree with you and apologize. We have already added it in section 2.7 under Materials and Methods.

recurrent cholesteatoma is time dependent, so the statistical analysis should consider time--> We understand and appreciate your concern regarding the consideration of time factor in our study. We want to assure you that we have taken time into account in our statistical analysis. Although our study was not prospective, we implemented measures to address this factor. We utilized appropriate logistic regression models that allow for handling time variables as covariates, ensuring that the potential effects of time on cholesteatoma recurrences were accounted for. Additionally, we employed robust statistical methods to ensure a thorough evaluation of this aspect. We hope these clarifications address your concerns, and we are open to providing further details if needed. We appreciate your feedback, which contributes to improving the quality of our work. Additionally, we have justified it in section 2.8 as follows: ‘’Understanding the importance of time in the development and recurrence of cholesteatoma, in our study, a statistical analysis was conducted using logistic regression to investigate the association between the occurrence of new recurrences and various parameters, including EAONO classification, pure tone audiometry (PTA), surgical technique utilized, type of reconstruction for aditus closure and ossicular chain, as well as findings from non-EPI-DWI MRI. While the study design was not prospective, measures were taken to control for the time factor in the statistical analysis, including the use of appropriate logistic regression models capable of handling time variables as covariates. Additionally, robust statistical methods were employed to ensure that the potential effects of time on cholesteatoma recurrences were adequately considered.’’

recurrent cholesteatoma is a clinical observation, imaging is useful for residual disease-> Thank you very much because we found this note very interesting. In fact, we have chosen to add it in order to clarify the information and avoid misunderstandings in the discussion as follows in line 406: ‘’In summary, to avoid misunderstandings, we would like to emphasize that the concept of recurrent cholesteatoma involves the reappearance of the disease, considering multiple factors as described in our study. Furthermore, the role of MRI and imaging diagnosis is noteworthy for monitoring and visualizing residual disease’’.

table 3: title of column 2 is p value, but coefficient values are reported--> You are absolutely right, and we have modified it in the caption by adding: "Table 3. Statistical analyses using logistic regression of possible factors associated with cholesteatoma recurrence using coefficient values."

Round 2

Reviewer 4 Report

Comments and Suggestions for Authors

table 3 is still incorrect, P values are still missing

"After the excision was performed, mesotympanic inspection with a 30 and 45-degree angled endoscope was carried out only in those patients with doubts regarding whether it had been complete." should be in materials and method, chapter surgical technique

Again, residual and recurrent disease should be distinguished, and MRI should be considered to check residual disease. In case off recurrent disease MRI is not useful unless intracranial complication are suspected.

Author Response

Thank you very much again for your comments, which we acknowledge and apologize for not having interpreted correctly according to your accurate criteria. We will address each point you mentioned.

1- Table 3 is still incorrect, P values are still missing--> We apologize for not understanding your suggestions correctly. We have recalculated all the statistical parameters of the study, making changes to some that you suggested were not correct in the logistic model. As a result, you will see new values in some cases. Fortunately, we did not have to modify all of them, and there were no changes regarding statistical significance. As you suggested, we have added both p-values and coefficients, along with their confidence intervals and Odds Ratios. We have mentioned this in both the text and tables, adding a new column specifically for this purpose. We hope this makes everything clear now.

2- After the excision was performed, mesotympanic inspection with a 30 and 45-degree angled endoscope was carried out only in those patients with doubts regarding whether it had been complete." should be in materials and method, chapter surgical technique---> Changed and added in materials and method.

3-Again, residual and recurrent disease should be distinguished, and MRI should be considered to check residual disease. In case off recurrent disease MRI is not useful unless intracranial complication are suspected--> We apologize for not understanding your previous message. As you suggest, we have changed all the statements in which we referred to the utility of MRI for detecting recurrence. We replaced 'recurrence' with 'residual cholesteatoma' in all contexts where we discussed imaging tests, making it clear, as you indicate, that recurrence is a clinical matter. The distinction is now clear. Additionally, we have added the interesting contribution you suggested by affirming that imaging tests are useful in case of recurrence with intracranial involvement, in line 475.
